# LLM-based Stock Market Trend Prediction

## Abstract

Investor sentiment, which is driven by 'intriguing factors' such as news articles and options volume, has been historically resistant to effective use in quantitative methods for predictive market analysis. The emerging science of large language models (LLMs), however, offers a potential solution to this problem. In this paper, we describe our initial experiments with a novel system which prompts available LLMs in a way which allows us to link responses with features in the otherwise more traditional quantitative methods. The results show high accuracy in predicting market moves. We describe the experiments and our initial thoughts about next steps in the paper.

## 1 Introduction

For years, quantitative analysts ("quants") have engaged in conventional statistical analysis of financial markets, largely driven by evident business incentives. The equities market, by its nature, is forward-looking and subject to the sway of numerous factors, primarily the sentiment held by potential investors. But how to 'read the market', owing to its volatility? Incorporating this sentiment into traditional statistical models is a key problem.

In this particular scenario, modern large language models (LLMs) seem to provide a potential solution. Our research paper introduces an initial investigation into employing generative techniques with a set of 'intriguing factors' to capture such sentiments for the purpose of predictive market analysis. The intriguing factors encompass moving averages, sentiments from news articles, options volume, as well as the interplay between supply and demand statistics.

Within the document, Section 2 presents an overview of Financial market analysis and outlines the potential utility of LLMs in this domain. Section 3 delves into the concept of moving averages and draws specific insights. Moving forward, Section 4 discusses the incorporation of news content along with its associated sentiment, highlighting their potential contributions to trend prediction. Within Section 5, the focus shifts to the volatility index and its correlation with options volume. The subsequent portion, Section 6, provides a concise introduction to the dependency tree, demand and supply chaining. In Section 7, we provide a brief account of our endeavors to effectively integrate these intriguing parameters using LLMs, along with the subsequent outcomes of this approach. The final section offers directions for future research and culminates in a concluding remark.

## 2 Financial Market Analysis

Significant investments orchestrated by major financial institutions are guided by proprietary financial models. Conversely, individual investors often rely on algorithmic trading or investment advice to gain insights into potential trends for specific company stocks. This paper underscores the potential benefits arising from the fusion of algorithmic trading logic, sentiment analysis utilizing LLMs, and a lesser-explored yet crucial heuristics: options derivatives, and demand supply factors which can exert influence over market volatility.

However, it's crucial to bear in mind that predicting trends within financial markets is essentially a calculated risk-taking endeavor. Such predictions remain susceptible to diverse deviations from established patterns at any given juncture. The inherent complexity and dynamic nature of financial

markets necessitate ongoing evaluation and a nuanced understanding of the inherent uncertainties.

LLMs inherently tap into extensive volumes of media reporting. Our initial findings corroborate that strategic prompts can introduce an input dimension that facilitates reasonable predictions of market movements. While no LLM can foresee the future or anticipate radical shifts in overall investor sentiment, they are adept at forecasting market behavior aligned with a specific corpus of media input. Notably, their predictions exhibit a level of granularity that holds actionable insights for investors.

Before getting into the technical details, we would want to go over two important techniques: Fundamental Analysis and Technical Analysis.

Fundamental analysis Wafi et al. (2015) seeks to assess stocks by gauging their inherent worth. Analysts conducting fundamental analysis delve into a range of factors, spanning the broader economic landscape and industry dynamics, down to the financial stability and managerial prowess of specific enterprises. Aspects such as earnings, expenditures, assets, and debts undergo meticulous examination within the purview of fundamental analysis.

Diverging from fundamental analysis, technical analysis Farias Nazário et al. (2017) involves traders seeking opportunities through the examination of statistical patterns, including shifts in a stock's price and trading volume. The central premise is that all relevant fundamentals are already incorporated into the price, rendering a detailed focus on them unnecessary. Technical analysts abstain from quantifying a security's inherent value; instead, they rely on stock charts to pinpoint configurations and trends that indicate the potential future behavior of a stock.

A perfect mix of Fundamental and technical analysis could serve as a benefactor in the stock market trend prediction.

## 2.1 MOTIVATION CHOICE OF PARAMETERS

Many investors use the average prices of the previous 10 days (p10), as well as p50 and p200, to predict the likely trend for the current day. We conducted a small test to see how accurate this method is, and it turned out to be 51 percent accurate. This looked like a good starting point for the next set of measurements ahead.

News related to a company, especially just before the market opens and the historical ones throughout the day, are very important and can impact how investors think. The challenge is that every aspect of investing depends on each other, creating a complex cycle that needs to find a balance point. Figuring out this balance point is quite tricky. To understand what triggers certain events accurately, we need to analyze news and its sentiment to get a better idea of market trends. Our next experiment involved combining average calculations with assessing news sentiments.

A fascinating part of financial markets is the ability to trade derivatives. These are called derivatives because their value comes from how the market reacts to a company's stock. However, the stock market is very sensitive to the movements of derivatives like options, swaps, and futures. Interestingly, the irregular movements of swaps indirectly triggered the 2008 financial crisis. We also conducted some small experiments in this area and found that the volume of options trading has a significant impact on various trading activities.

A dependency tree helps assess what is the supply chain health of a company. This tree when analyzed thoroughly can estimate the supply heuristics. One can determine the demand driving (of a company) figures and establish what could be the probable trend of its stock. Getting this handy can connect several dots together. This way the dependency, demand, and supply factor was considered as another interesting parameter. A dependency tree is a tool that assists in evaluating the overall

health of a company's supply chain. By closely examining this tree, it becomes possible to gauge the supply-related patterns. With a proper Demand factor analysis and supply statistics, the prediction of potential stock trends becomes more accurate. Having access to this information allows for the linking of various pieces of information. This is why the interaction between dependency, demand, and supply factors was regarded as an additional intriguing parameter.

## 3 N-DAY AND 'MOVING AVERAGES'

Simple moving averages serve as indicators that aid in evaluating a stock's trajectory by averaging its daily prices across a set time-frame. Signals to buy or sell are triggered when a shorter-term moving average intersects a longer-term onede Souza et al. (2018). Support and resistance rely on historical price data. Support zones are characterized by instances where buyers have intervened previously, while resistance zones indicate points where sellers have hindered price advancement. Practitioners typically seek buying opportunities at support levels and selling opportunities at resistance levels.

Trend lines, akin to support and resistance, furnish specific entry and exit reference points. However, they differ by extrapolating from past trading patterns. These lines are particularly valuable for stocks reaching new highs or lows, where past price data might be limited. We conducted a series of experiments to explore the impact of three different time measurements on market trends. Our experimentation journey commenced with analyzing the influence of a single measure, p10 (representing a 10-day time period). Subsequently, we progressed to evaluating the combined effects of p10 and p50, followed by p10, p50, and p200. Our investigation continued by including p365, and ultimately extending to incorporate p500.

During these experiments, we observed a distinct pattern in the accuracy of our predictions. As we introduced more time measurements, the accuracy initially improved, peaking when considering the combination of p10, p50, and p200. However, beyond this point, the accuracy began to decline. This phenomenon aptly highlights the specific time-period range that wields a significant influence over a particular stock's performance.

The graph provided below visually encapsulates this trend, showcasing the interplay between the number of time measurements and the corresponding prediction accuracy. This analysis led us to the conclusion that the most effective consecutive combination of time periods was p(10+50+200), which struck a balance between capturing meaningful patterns and avoiding over-complication. This selection emerged as the optimal configuration for understanding and predicting market trends for the stock in question.

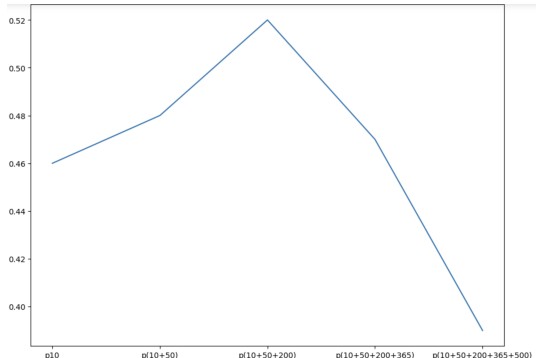

Figure 1: Moving Averages

## 4 NEWS - WHAT TRIGGERS WHAT

We aimed to investigate the predictability of daily trends and adopted a methodology primarily centered around understanding causal relationships. Our approach involved collecting intra-day stock price movements, sourcing pertinent news data, and then averaging stock prices based on the temporal boundaries of news releases. The objective was to ascertain how individual news items influenced stock prices until the emergence of subsequent related news pieces. To link news items with sentiments, we associated a sentiment score (+1 for positive, -1 for negative, and 0 for neutral) with each news feed. It's worth noting that our approach did not incorporate the concept of p(10+50+200), which represents certain time periods.

This framework allowed for informed speculation about instantaneous trends. However, the accuracy of our predictions was moderate, hovering between 0.70 and 0.72. For the news feed streaming, we utilized StockNewsAPIs, which internally harnessed LSTMs to analyze news sentiment based on their subject matter and descriptions.

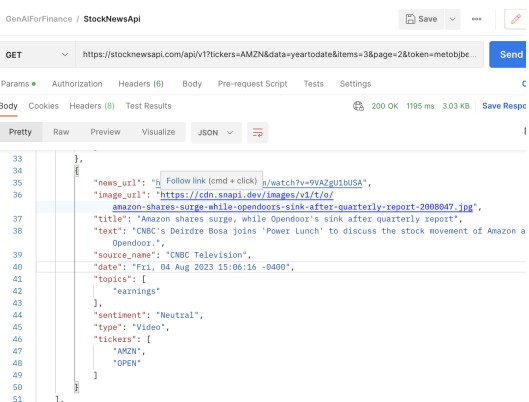

Figure 2: StockNewsApi Response Format Link

We endeavored to enhance prediction accuracy by fine-tuning our approach, and applying different sentiment weightings to news feeds (a different API that utilized LLMs internally). This approach helped improve a little, achieving an accuracy of approximately 0.73. It facilitated the machine in recognizing correlations between news items and stock price fluctuations of varying degrees.

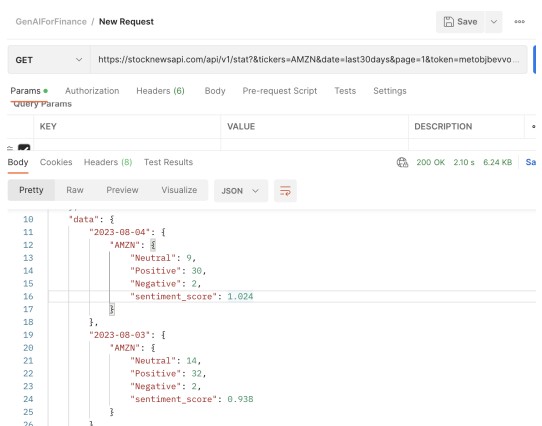

Figure 3: StockNewsApi Sentiment Response Format

Nevertheless, this form of analysis exhibited limited confidence due to instances where the market responded differently to positively influencing news feeds. Delays in market responses were occasionally observed, potentially due to cumulative factors beyond our control. Approaches like this necessitate sensitivity to news content, stock price dynamics, and other interdependent variables. They should be adaptive, constantly learning, and executed at short intervals.

We concluded that predicting the trend for the 'n+1'th day using data from preceding 'n' days could be more valuable. To achieve this, we combined both the p(10+50+200) concept and news data. Our evaluation involved predicting a test set with news data limited to the pre-market news window. The associated sentiments were again categorized as +1, -1, or 0. This modified approach substantially improved accuracy, achieving a level of 0.78.

## 5 Derivatives influencing the market

The Volatility Index (VIX) serves as an indicator for anticipating market volatility. VIX is the annualized implied volatility of a hypothetical S and P 500 stock option with 30 days to expiration. The price of this option is based on the prices of near-term S and P 500 options traded on CBOE. It can help investors estimate how much the S and P 500 Index will fluctuate in the next 30 days.

$$\sigma^2 = \frac{2}{T} \sum_i \frac{\Delta K_i}{K_i^2} e^{RT} Q(K_i) - \frac{1}{T} \left[ \frac{F}{K_0} - 1 \right]^2 \quad (1)$$

**Where**

| | | | |
|---|---|---|---|
| $\sigma$ | $VIX/100 \Rightarrow VIX = \sigma \times 100$ | $\Delta K_i$ | Interval between strike prices – half the difference between the strike on either side of $K_i$ |
| $T$ | Time to expiration | | $\Delta K_i = \frac{K_{i+1} - K_{i-1}}{2}$ |
| $F$ | Forward index level derived from index option prices | | |
| $K_0$ | First strike below the forward index level, F | $R$ | Risk-free interest rate to expiration |
| $K_i$ | Strike price of $i^{th}$ out-of-the-money option; a call if $K_i > K_0$ and a put if $K_i < K_0$; both put and call if $K_i = K_0$. | $Q(K_i)$ | The midpoint of the bid-ask spread for each option with strike $K_i$. |

Figure 4: Formula to calculate VIX index

Strike price is the price at which the underlying security in an options contract can be exercised, and is directly related to options volume. The VIX's computation hinges on options trading volume, particularly the ratio of put options volume to call options volume Li (2021). Another interesting fact is that VIX owing to its volatility, has seen depicting the trend in market appropriately. The graph below can explain this fact in a clearer way. Fluctuating rise can hint towards a downtrend and vice versa.

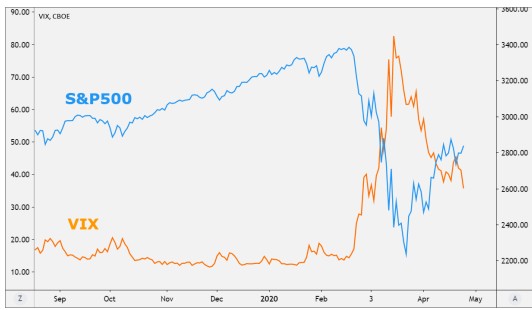

Figure 5: VIX Index

With this in mind, we explored the utility of the options data, considering the insight that "today's market trend has correlation with the previous day's options trading volume. Exchange (2022)"

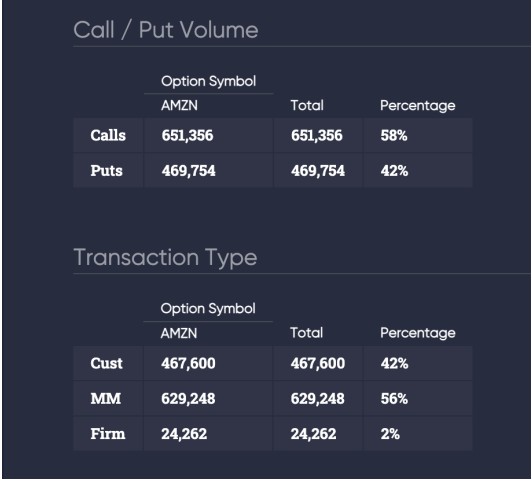

Figure 6: Options trading Volume info for AMZN on August 9th that could be useful to predict the next day's trend

Inculcating this, we were able to achieve accuracy of 0.92. [Options volume Datasource]

## 6 THE STORY OF DEPENDENCIES, DEMAND AND SUPPLY

According to the studies by Harvard Business School Barnett (1988), the process of creating a comprehensive market forecast involves different steps:

1. Market Definition: Investors should clearly define the scope and boundaries of the market under consideration.
2. Supply chain analysis: Analyzing the supply chain statistics of the company, and its situational analysis that would essentially predict and understand the health of related companies within the "supply" bracket.
3. Value Segmentation: Breaking down the overall industry-value demand into its primary components or segments and predicting the factors influencing demand within each segment and anticipating how they might evolve over time.
4. Sensitivity Analysis: Assess the most crucial assumptions and potential risks associated with the baseline forecast through sensitivity analyses. This helps to understand the extent of their impact and to evaluate potential risks to the forecasted outcomes.

This process is far from straightforward, and the current capabilities of LLMs are not optimally suited for its efficient execution. There remains ample room for improvement in this area. Taking a step further, we endeavored to construct intricate supply-chain-demand dependency graphs for specific companies that encompassed a network of interconnected entities. For instance, Company A relied on the operations of companies B, C, and D. Furthermore, Company B's functioning was contingent upon companies E and F, and similar dependencies existed for companies C and D. This structure can be envisioned as an n-ary tree. The branches of this graph primarily consist of natural resources and their various compositions that facilitate production in the higher-tier company nodes.

When the supply chain is operating smoothly, production typically proceeds without issues. Conversely, disruptions (such as a Chip Shortage) coupled with high demand can trigger price increases. As a result, the assessment of demand drivers, supply chain metrics, and dependency

relationships assumes paramount importance in accurate market forecasting.

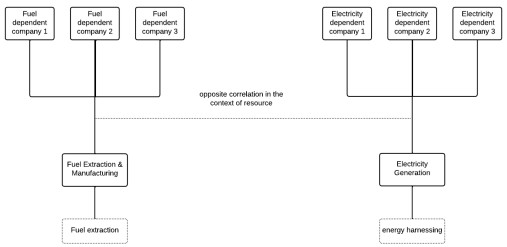

Figure 7: Dependency Graphs

These trees can be used to depict related correlations as well. A significant decrease in the fuel prices can bring down the demand on fuel powered automobile industries, thus their prices might decrease. If the fuel import faces issues at the port and is not able to make it, it could put the fuel on high demand, thus increasing prices of fuel as well as the companies depending on this fuel. This price rise might start making people lean towards electricity dependent companies. Such dependencies have to be considered while making decisions. The following diagram is very well known in the context of finance and is termed the Demand—Supply—price graph.

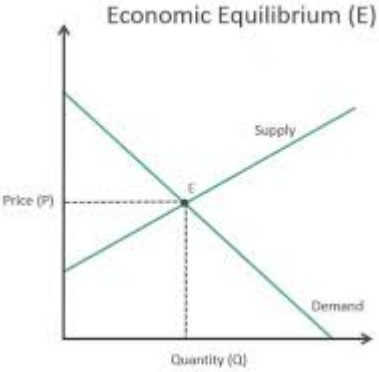

Figure 8: Equilibrium point - Demand Supply analysis

The intersection of the demand and supply curves is where market equilibrium occurs Fernando (2023). This is when the amount people want to buy matches the amount available for sale at a specific price. This balanced price is also referred to as the market-clearing price. If either demand or supply changes, the curves shift, causing adjustments in both the equilibrium price and quantity.

By instructing the Language Model (LLM) to analyze company reports and comprehend the dependency graph, as well as understand demand and supply semantics and supply chain logs, the accuracy of predicting trends improved. We provided company-related documents to evaluate the supply chain and demand-driving factors. Utilizing an interconnected n-ary tree representation and considering historical data from related sectors, we calculated an overall performance factor. This factor displayed a strong correlation with the company's stock trend on a specific day. When combined with the previously mentioned parameters, these insights could enhance trend predictions. This entire process can be automated by training LLMs to systematically consider all relevant parameters, draw inferences, and predict trends for any given day.

## 7 RESULTS AND OBSERVATIONS

We aggregated all the previously mentioned parameters and employed a TensorFlow machine learning model to observe how the model behaved. The accuracy showed incremental improvement, starting from 0.52 (p10+p50+p200) and progressing to 0.78 (news sentiment), 0.92 (options volume), and ultimately reaching 0.95 (dependency, demand, and supply - DDS using LLMs).

However, the process of generating the DDS factor was executed manually. This involved tasks such as sourcing information, training the dependency graph, and refining scraped data. Notably, we bypassed the use of APIs and directly engaged the Language Models (LLMs) to fetch and interact with the necessary details which were later fed to the model.

We carried out several additional experiments involving different companies (such as NVIDIA and Coinbase) along with their associated datasets, which included historical stock trends, sentiments from news articles, options trading volumes, and a demand supply dependency sentiment. These experiments were conducted in batches (and in different combinations) to also assess their performance on older datasets for validation purposes.

Throughout our analysis, we made several noteworthy observations that can be summarized as follows:

1. Regarding NVDA, the inclusion of news sentiment analysis, options trading volumes, and the supply and demand dependency graph contributed significantly to the accuracy of predictions. Solely relying on moving averages was insufficient in accurately pinpointing key turning points, particularly the instance where NVDA's stock price began a substantial and sustained upward trajectory. The introduction of news sentiment analysis notably enhanced predictive accuracy. Options trading volume also played a crucial role in bolstering confidence in predictions. Additionally, the dependency graph depicting supply and demand dynamics acted as a stabilizing factor, rectifying potential inaccuracies in predictions. Consequently, the achieved accuracy level reached 0.94.

2. Conversely, in the case of Coinbase, predictions based solely on news sentiment analysis exhibited suboptimal performance. In some instances, there was an adverse correlation, indicative of situations where investor sentiment shifted unexpectedly. By incorporating options trading volumes, the dependency graph, and foundational moving averages, the prediction accuracy substantially improved. The overall accuracy approached 0.91.

Moving forward, our subsequent objective revolves around determining the extent of the upward trend's percentage increase or the extent of downward trend's percentage decrease. This determination will be influenced by the relative weights assigned to the distinct categories of data utilized in our analysis. These weightings are contextually dependent and may undergo daily adjustments. To address this, we intend to leverage Large Language Models (LLMs) to facilitate this process as well.

## 8 CONCLUSION

We were pleased with the accuracy of these initial predictions, but recognize there are a number of threats to validity which must be addressed moving forward. Obviously, the scale of the validation can and should be increased, and conducted across more sectors. In addition, whatever the scale of data being used, we would also like to conduct additional cross-validation activities, which we have not yet pursued as they are resource intensive; now that the pilot activities have panned out so well, this is something to circle back to and evaluate the breadth and stability of our results.

One caution to keep in mind is our use of open ChatGPT services. Our experiments are very much at the mercy of the LLMs as trained out of our control, and while the services are truly breathtaking in scope and power, they are also clearly a moving target, and one can envision the training sets evolving over time (or potentially polluted in ways that are becoming evident in some current applications.)

We also plan to compute the alteration in the percentage shift of the stock's price, as elaborated earlier. This computation has the potential to enhance the overall understanding and enable investors to make more informed and precise decisions.

Ultimately of course application of this approach would require additional investment into our software apparatus in order to enable continuous monitoring of the input data. Our work in this paper is based on snapshots processed monolithically, and thus they offer great hindsight. In practice, one would certainly employ an adaptive version of these algorithms in order to rapidly adjust to fast-moving market conditions and sentiment.

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
