# OpenReview forum: "LLM-based Stock Market Trend Prediction"
_ICLR.cc/2024/Conference — Submitted to ICLR 2024_

### Official Review · Reviewer_fQZv · 2023-10-13

**Soundness:** 1 poor
**Presentation:** 1 poor
**Contribution:** 1 poor
**Rating:** 1
**Confidence:** 5

**Summary:**

The paper delves into the topic of stock market trend prediction but it is a clear reject.

**Strengths:**

See weaknesses

**Weaknesses:**

* The paper lacks a clear structure, making it difficult to identify sections dedicated to previous work, the proposed method, and experiment settings. This ambiguity necessitates extra effort from the readers to discern these distinctions.
* The literature review appears insufficient. The absence of a dedicated "related work" section and the lack of references from the domain are evident. The authors can consider papers [A systematic review of fundamental and technical analysis of stock market predictions] and [Neural network techniques for financial performance prediction: integrating fundamental and technical analysis] to make better literature review.
* There is ambiguity surrounding the proposed method. It remains unclear what exact approach the authors have proposed.
* The use of ChatGPT services, presumably leveraging LLMs, is revealed only at the end. This critical information should be upfront, as the experiments' outcomes heavily rely on ChatGPT and not even  involving training LLM themselves.
* It's unclear which dataset was employed, its preprocessing steps, and the data splits used.
In Section 3, the link between the established conclusion (regarding the most effective consecutive combination of time periods) and the underlying motivation to use LLM remains ambiguous.
* The method of computing accuracy in Section 4 is not explained. Given the mentioned accuracy range 0.7-0.72, the efficacy of the proposed approach seems questionable. The marginal improvement in accuracy using LLM raises questions about its necessity for this task. (From 0.72 to 0.73). Section 4 also aims to delve into causal relationships, but it lacks clarity. The results also seem misaligned with the stated objective.
* Figure 1 lacks a clear y-axis label and accompanying caption to interpret its contents. The relevance and contribution of Figures 2 and 3 are unclear, as they do not appear to offer useful insights. Some expressions, such as "We were pleased with the accuracy," are unorthodox for academic literature.
* Section 5, discussing the Volatility Index (VIX), does not evidently connect with the paper's primary methodology or objectives. The experiments do not provide clear evidence supporting the advantages of using LLM for the stated task.
* The paper lacks baseline comparisons to evaluate the proposed method's effectiveness compared other established methods.
* The paper's contributions is unclear and remain questionable. The alignment between the experiments and the paper's motivation is not evident.

**Questions:**

See weaknesses

---

### Official Review · Reviewer_Tvq8 · 2023-10-23

**Soundness:** 1 poor
**Presentation:** 1 poor
**Contribution:** 1 poor
**Rating:** 3
**Confidence:** 4

**Summary:**

Paper describes some initial investigations in using LLMs for trend prediction in finance.

**Strengths:**

1. Has some experimental metrics.

**Weaknesses:**

1. Does not describe specific procedures, prompts, or other steps taken toward improving LLM performance.
2. Does not cite related work (such as https://arxiv.org/abs/2303.17564) - there are several more. This is just one example.
3. Paper is poorly written jumping from one concept to another with very sparse introductions and less focus on details.

**Questions:**

1. Can authors describe their processes in more detail?
2. Can authors provide performance over multiple time periods and multiple assets (preferably asset classes).

---

### Official Review · Reviewer_Tge5 · 2023-10-30

**Soundness:** 1 poor
**Presentation:** 1 poor
**Contribution:** 1 poor
**Rating:** 1
**Confidence:** 5

**Summary:**

This paper describes an approach to predicting stock market trend using LLMs. The authors demonstrate that by leveraging LLM they were able to achieve an accuracy of 95% on the options volume dataset. However, the authors seem to be lacking a solid machine learning background, and therefore I have doubts about the correctness of the reported accuracy metrics. In addition, the paper is poorly written and hard to read.

**Strengths:**

The author reported a very intriguing 95% of accuracy on the stock market trend prediction task, which is significantly higher than what people usually expect on the highly noisy stock data.

**Weaknesses:**

- This paper is very poorly written. There are many format errors, and only a very limited number of related works are referenced. The figures are also poorly made.
- The technical contribution of this paper is very marginal, since it is a naive incorporation of the LLM.

**Questions:**

What is the start and end date of the training dataset? How is it split into training/validation/test sets? How are news articles and price data aligned to ensure that there is no leak? These details were not clearly written in the paper.

---

### Official Review · Reviewer_m81M · 2023-11-03

**Soundness:** 1 poor
**Presentation:** 1 poor
**Contribution:** 1 poor
**Rating:** 1
**Confidence:** 5

**Summary:**

This paper presents a decent overview of the types of factors that drive stock prices, including technical analysis based on price and volume data, natural language processing on news data to extract sentiment and flows implied by daily options trading.  The main contribution of the paper appears to be the use of large language models (LLMs) to extract a supply and demand dependency graph for companies based on company reports (e.g. 10qs).  The output of the dependency graphs are then combined with technical indicators (e.g. moving averages), news sentiment and volatility indicators to predict future stocks prices.

**Strengths:**

The theme behind this paper is interesting: how can LLMs be used to improve investing and, specifically, what types of complex analysis can LLMs perform given a set of company reports.  And in general, I think the idea of combining fundamentally driven analysis with technical analysis is a solid way to think about systematic investing.

**Weaknesses:**

The paper was written in a manner that made it very difficult to parse out the actual contribution.  It helps to be explicit - i.e. list contributions as bullets at the end of the introduction.

The paper is applied/empirical in nature. However, I do not think a researcher, even one intimately familiar with systematic equities investing, could pick up this paper and recreate any of the experiments.

In addition, it is nearly impossible to interpret the empirical findings.  The improvement in accuracy from 0.52 to 0.78 seems impressive, but no benchmark was considered.

**Questions:**

Too many clarification questions to list here, so I'll include just a few:
-In the N-Day Moving Averages section:
  - what data was considered?  daily, hourly, tick level,...? is the average a weighted average?
  - how were the different averages combine?
  - how were the averages evaluated / what is the y-axis in Figure 1?
-In the News Sentiment section:
  - where does the sentiment come from?
  - what accuracy measure is used? which LLM was used?
-In the supply and demand dependencies section:
 - how were the LLMs used here?  What questions/prompts were used?  how did you evaluate the graphs?
-In the results section:
 - what was the final model?
 - what are you predicting?
 - what data was used for training? what data was used for evaluation?
 - what is performance measure?

---

### Meta-Review · Area_Chair_QGVp · 2023-12-09

**Metareview:**

The paper aims to leverage LLMs for predicting trends in stock market. The reviewers have found that paper is not well written and there are many ambiguities. Additionally, the experiments are not very reproducible and the technical idea relies on a naive incorporation of LLMs. Based on unanimous reviewer feedback, the decision is reject.

**Justification For Why Not Higher Score:**

N/A

**Justification For Why Not Lower Score:**

N/A

---

### Decision · Program_Chairs · 2024-01-16

Reject